# Anaemia and Nutritional Status during HIV and Helminth Coinfection among Adults in South Africa

**DOI:** 10.3390/nu14234970

**Published:** 2022-11-23

**Authors:** Miranda N. Mpaka-Mbatha, Pragalathan Naidoo, Md Mazharul Islam, Ravesh Singh, Zilungile L. Mkhize-Kwitshana

**Affiliations:** 1Department of Biomedical Sciences, Faculty of Natural Sciences, Mangosuthu University of Technology, Umlazi, Durban 4031, South Africa; 2Department of Medical Microbiology, School of Laboratory Medicine and Medical Sciences, College of Health Sciences, Nelson R. Mandela Medical School Campus, University of KwaZulu-Natal, Durban 4001, South Africa; 3Division of Research Capacity Development, South African Medical Research Council, Tygerberg, Cape Town 7505, South Africa; 4Department of Animal Resources, Ministry of Municipality, Doha P.O. Box 35081, Qatar; 5Department of Medical Microbiology, School of Laboratory Medicine and Medical Sciences, College of Health Sciences, Howard College Campus, University of KwaZulu-Natal, Durban 4041, South Africa

**Keywords:** HIV, helminths, coinfection, anaemia, malnutrition

## Abstract

Sub-Saharan Africa is burdened with helminthiasis and HIV/AIDS, and there is a significant overlap between these infections. However, little is known about the extent of anaemia and malnutrition in HIV/AIDS and helminth coinfected adults. The study investigated the anaemia profiles and nutritional status of HIV and helminth coinfected adult South Africans. Stool samples were collected from participants (N = 414) for parasite detection using the Kato–Katz and Mini Parasep^®^ SF techniques. Blood was collected to determine participants’ HIV status, micro- and macronutrients, haematological parameters, and *Ascaris lumbricoides*-specific IgE and IgG4 levels. Thereafter, participants were stratified into single infection (HIV or helminths), coinfection, and uninfected controls (no HIV and helminth) groups. The majority (74.9%) of participants had CD4 counts of >500 cells/μL, indicating no significant immunosupression. The coinfected group had an overall anaemia prevalence of 16.9%, which was lower than that of the HIV-infected group (44.6%) and higher than helminth infected group (15.4%). Overall helminth prevalence was 33%, with *Ascaris lumbricoides* being the most prevalent. The coinfected group also had lower vitamin A (*p* = 0.0107), calcium (*p* = 0.0002), and albumin (*p* < 0.0001) levels compared to HIV/helminth uninfected controls. Unexpectedly, the coinfected group had the highest serum iron levels, followed by the helminth-infected and control groups, both of which had similar iron levels, and finally, the HIV-infected group, which had the lowest iron levels (*p* = 0.04). Coinfected adults may be prone to micronutrient deficiency and anaemia. Further research and intervention programmes are required in this neglected field.

## 1. Introduction

The human immunodeficiency virus (HIV) is a lentivirus that causes the acquired immunodeficiency syndrome (AIDS) [1]. HIV enters the cells through the CD4 molecule using the CCR5 and/or CXCR4 coreceptors. The virus, therefore, replicates in all CD4 expressing cells such as T cells and monocytes, ultimately leading to their depletion and immune system impairment [2]. It is estimated that 40 million people are infected with HIV worldwide [3]. To date, this global pandemic has claimed over 36.3 million lives [4]. Although the management of HIV has improved over the years due to increasing access to prevention, diagnosis, and treatment interventions, the World Health Organization (WHO) reported 34.8 million people living with HIV in 2021, with the majority (67%) living in Africa [3]. In 2020, it was estimated that 1.5 million people were newly infected with HIV-1 in Africa [3]. Approximately 25% of all new infections were reported in Southern Africa, making it the worst affected region on the continent [5]. In the absence of a cure or vaccine, the WHO has recommended that all people living with HIV be provided with lifelong antiretroviral treatment (ART) [4].

Approximately 1.5 billion or 24% of the world’s population is estimated to be infected with intestinal helminths [6]. The WHO has listed these infections as part of the neglected tropical diseases [7]. Helminth infections continue to overburden those living in poor and deprived communities, particularly those living in the tropical and sub-tropical regions with poor sanitation and lack of a clean water supply [8]. The major helminthic species of public health significance in Sub-Saharan Africa (SSA) are hookworms (*Necator americanus* or *Ancylostoma duodenale*), *Ascaris lumbricoides*, *Trichuris trichiura*, *Schistosoma haematobium*, and *Schistosoma mansoni*. SSA also has the highest malnutrition prevalence estimates in the world, with 23.2% of the population suffering from the condition [9]. A widespread and geographic overlap between HIV and intestinal helminths has been reported in SSA, including South Africa (SA) [8,10].

The helminths and HIV infection overlap may exacerbate anaemia since both infections are associated with nutritional deficiencies [11], which results in a complex manifestation (HIV, helminthiasis, and malnutrition). HIV infection impairs the individual’s metabolic ability to absorb, store, and utilize nutrients, resulting in nutrient deficiencies, compromised immunity, and an increased risk of developing infectious diseases [12,13]. Inadequate food intake, combined with malabsorption, lead to further progression of the HIV infection to AIDS [13]. Severe malnutrition is a significant predictor of AIDS-related morbidity and mortality [13]. Iron deficiency is the most common nutritional disorder globally and it is highly prevalent in geographical areas that favour the manifestation of HIV and helminth infections [14]. HIV-helminth coinfected individuals may be prone to suppressed immune responses and are at a greater risk of acquiring other infections since nutrition is important for a strong immune system [15]. Moreover, HIV may lead to bone marrow failure, which reduces the production of red blood cells, leading to anaemia [16]. Malnutrition worsens the disease status of coinfected individuals by accelerating the progression of HIV infection to AIDS [12]. KwaZulu-Natal (KZN) has been reported as the province in SA with a higher prevalence of both HIV and helminth [11,17]. Recently, a study reported more than a 30% prevalence of helminth infections among adults in KZN [11]. The tropical weather in the province influences the high prevalence of soil-transmitted helminths and *Schistosoma* [11,18]. There are currently insufficient data on the prevalence of malnutrition and anaemia among people living with HIV in SA, including KZN. Therefore, the current study aimed at investigating the anaemia and nutritional status of adults with HIV and helminth coinfection in a South African adult population residing in KZN.

## 2. Materials and Methods

### 2.1. Study Area and Recruitment of Study Participants

A cross sectional study was undertaken between March 2020 to May 2021. A participant sample size of 318 was shown to have 80% power to show differences in groups with a 95% confidence interval based on the 29.2% helminth prevalence reported in KZN in an adult population [19]. Our study, however, included more (N = 414) participants to cater for loss to follow up. The study area is situated within a subdistrict of the eThekwini district, a peri-urban area located in the south of Durban, KZN province, SA, which has eleven primary health care clinics (PHC). Water sources varied from taps in the homes and outside of the homes, communal taps, and other sources (rivers, boreholes, and municipal water delivery trucks). Likewise, toilet use also varied from pit toilets and flush toilets connected or not connected to sewerage [20]. Of the eleven clinics, five were selected as the study sites based on the availability of HIV counselling and testing (HCT) services. 

Adult participants (aged ≥18 years) were purposively selected from HCT sites. The study included clinic patients, and other participants who accompanied patients to the clinic and community members who were recruited by friends and family members attending the selected clinics. The details of participant selection, inclusion and exclusion criteria, and their demographic profile were described previously [20]. Most of the participants who were recruited by friends and family members were not clinic attendees and had not been tested for HIV. Therefore, all the participants were tested and those who had already been tested by the clinics were retested during the study experiments to further confirm their HIV status.

### 2.2. Sample Collection

Whole blood was collected from all participants using the Vacuette^®^ EDTA and serum separator tubes (SST) for haematological, virological, and biochemical parameters and *Ascaris lumbricoides*-specific IgE and IgG4 analysis. Stool samples were also collected for parasite detection. 

### 2.3. Parasite Detection

All stool samples were prepared and processed for coproscopy using the Kato–Katz (Sterlitech: Auburn, Washington, DC, USA) quantification technique and the Mini Parasep^®^SF modified faecal parasite concentrator technique (Apacor Ltd., Wokingham, UK). To improve the sensitivity of coproscopy, serum samples were sent to the Allergology and Clinical Immunology unit diagnostic research laboratory, Groote Schuur Hospital, University of Cape Town for the serological detection of *Ascaris lumbricoides*-specific IgE (>0.35 kU/L) and IgG4 (>0.15 kU/L) levels using the Phadia™ 200 instrument (ThermoFisher Scientific: Phadia AB, Uppsala, Sweden). Previously published studies used serological detection of *Ascaris lumbricoides* IgE and IgG4 to improve coproscopy sensitivity [11,21,22]. Participants with *Ascaris lumbricoides*-specific IgE levels > 0.35 kU/L and IgG4 levels > 0.15 kU/L indicated infection [11,21,22]. Participants were stratified into four groups based on their HIV and helminth infection status as follows: (i) HIV and helminth uninfected controls; (ii) helminth-infected; (iii) HIV-infected; and (iv) coinfected (HIV and helminth infected). The description of the of the parasitic findings was presented elsewhere [20].

### 2.4. Detection of HIV Status, Viral Loads, and CD4+ Counts

Using serum samples, HIV status of the participants was confirmed using an Alere Determine^TM^ HIV-1/2 Ag/Ab Combo rapid test Orgenics Ltd., Yavne, Israel). Inconclusive results were confirmed by using the ICT HIV-1/2 Ag/Ab test kit (ICT Diagnostics, Cape Town, South Africa). HIV viral load and CD4+ counts were analysed at Neuberg Global Laboratory, a South African National Accreditation Standards (SANAS) accredited diagnostic laboratory, located in Durban, South Africa. 

### 2.5. Analysis of Anaemia and Nutritional Status

Anaemia was determined by both haemoglobin (Hb) and iron parameters. For the assessment of nutritional status, both anthropometry and biochemical analysis of macro and micronutrients were determined. Blood samples that were collected from the participants were taken to Neuberg Global Laboratory, a SANAS accredited diagnostic laboratory for haematological and biochemical analysis (Table 1). Hb levels and mean corpuscular volume (MCV) were detected using a Sysmex XN-1000™ Haematology Analyzer (Sysmex, Europe). Biochemical parameters were detected using a UniCel DxC 600i Chemistry Analyser (Beckman Coulter: Brea, CA, USA).

### 2.6. Data Analysis

The laboratory data were captured in a Microsoft Excel 2016 spreadsheet, and statistical analyses were performed using the STATA/IC-17 (STATA Corp LLC, Lakeway Drive, TX, USA) and GraphPad Prism 5 (GraphPad Software, Inc., San Diego, CA, USA) statistical software packages. Descriptive analysis for clinical parameters were presented as the number (n) and percentage (%), with a 95% confidence interval (CI). Univariate analysis was done by the one-way analysis of variance (ANOVA) Bartlett’s equal-variance test to identify the association between HIV-infected, helminth-infected, HIV/helminth coinfection, and uninfected control groups with anaemia and nutrition status. A *p* < 0.05 was considered statistically significant.

## 3. Results

### 3.1. General and Clinical Profiles of Participants

The mean age of the 414 participants was 41 years, where the majority (51.2%) were between the ages of 35–59 years, female (67.0%), and overweight/obese (60.9%). Overall, 52% of the participants were HIV infected, 33.0% were infected with helminths and 15.0% were coinfected with HIV and helminths [20] An HIV viral load of ≤50 copies/mL is classified as undetectable by the WHO [23]. In the present study, the classification of HIV viral load was based on the laboratory method detection limit; of ≤20 copies/mL hence all such levels were classified as undetectable(Table 2). CD4 counts were also categorized as per WHO guidelines as follows: (i) a CD4 count of <200 cells/μL indicates severe immunosuppression; (ii) a CD4 count of 200–349 cells/μL indicates advanced immunosuppression; (iii) 350–499 cells/μL indicating mild immunosuppression; and (iv) a CD4 count of ≥500 cells/μL indicates no significant immunosuppression [24]. In the present study, the majority (74.9%) of participants had CD4 counts of >500 cells/μL and only 5.8% had CD4 counts below 200 cells/μL (Table 2). A proportion of the participants, 47 of 414, were taking supplements; however, none were iron supplements. 

Overall anaemia as defined by Hb levels (Hb < 12g/dL) was 31.4%. Mild and moderate anaemia was more common among females (111/276, 40.2%) compared to males (27/138, 19.6%). Four of the 276 females (1.45%) had severe anaemia while none of the males had severe anaemia. The majority of the participants had a high (83.36%) MCV. Regarding malnutrition status, of the 414 study participants, 31.2% had low iron levels, 28.0% had low ferritin levels, 3.6% had low transferrin, 46.4% had low transferrin saturation levels, 6.5% had low calcium levels, 5.8% had low vitamin A levels, 0.2% had low zinc levels, 1.9% had low magnesium levels, 7.7% had low phosphate levels, 8.0% had low albumin levels, and 0.2% had low total protein levels (Table 2). 

### 3.2. Anaemia Status

#### 3.2.1. Frequency of Anaemia by Haemoglobin Analysis

The coinfected group had an overall anaemia prevalence of 16.9%, while the HIV and helminth singly infected groups had a prevalence of 15.4% and 44.6%, respectively. Approximately 23% of HIV/helminth uninfected controls were anaemic (Figure 1).

#### 3.2.2. Haematological and Biochemical Analysis of Anaemia and Nutritional Status

Comparisons between the coinfected and singly infected or uninfected groups were made to ascertain whether coinfection worsens anaemia and nutrition. Data presented were not adjusted for participants’ age, gender, BMI, and vitamin and mineral supplements intake. The coinfected group had the highest serum iron levels, followed by the helminth-infected and HIV/helminth uninfected control groups, both of which had similar iron levels, and finally, the HIV-infected group, which had the lowest iron level (*p* = 0.0410); the coinfected group had significantly higher iron levels compared to the HIV infected group (*p* = 0.0066). The HIV/helminth uninfected control group had the highest Hb levels, followed by the helminth-infected and coinfected groups, both of which had similar Hb levels, and finally, the HIV-infected group, which had the lowest iron level (*p* = 0.0332); expectedly, the HIV/helminth uninfected control group had significantly higher Hb levels compared to the HIV-infected group (*p* = 0.0032). No significant differences were noted for transferrin, transferrin saturation, ferritin, and zinc between the coinfected and singly infected groups (Figure 2). 

### 3.3. Nutritional Status

#### 3.3.1. BMI Distribution and Nutritional Profiles in the Study Population

Figure 3 highlights the anthropometric measurements [underweight (<18.5 kg/m^2^), normal weight (18.5–24.9 kg/m^2^), overweight (25.0–29.9 kg/m^2^), and obese (≥30.0)] in the study population categorised based on infection status. In the coinfected group, 3.0% of participants were underweight, 47% were normal weight, 19.0% were overweight, and 31.0% were obese. In the HIV-infected group, 8.0% of participants were underweight, 32.0% normal weight, 27.0% were overweight, and 33.0% were obese. In the helminth-infected group, 7.0% of participants were underweight, 35.0% normal weight, 28.0% were overweight, and 31.0% were obese.

#### 3.3.2. Biochemical Analysis of Nutritional Status

Data presented in Figure 4 were not adjusted for participants’ age, gender, BMI, and vitamin and mineral supplements intake. The coinfected group had significantly lower vitamin A (*p* = 0.0030), calcium (*p* = 0.0002), and albumin (*p* = 0.0008) levels compared to the HIV/helminth uninfected control group. In addition, the coinfected group had significantly lower calcium (*p* = 0.0362) and albumin (*p* = 0.0075) levels when compared to the helminth singly infected group. No significant differences were noted for total protein, magnesium, and phosphate (Figure 4).

### 3.4. Multivariate Regression Analysis

Data for multivariate regression analysis associating participants’ micro and macro-nutrient levels and Hb levels with their infection status are presented in Table 3. All data were adjusted for participants’ age, gender, BMI, and vitamin and mineral supplements intake. The coinfected group had significantly lower calcium levels (β = −0.05 mmo/L, *p* = 0.003), albumin levels (β = −2.13 g/L, *p* = 0.001), and vitamin A levels (β = −91.44 μg/L, *p* = 0.40) compared to the HIV/helminth uninfected controls. The HIV-infected group had significantly lower calcium levels (β = −0.04 mmo/L, *p* = 0.005) and albumin levels (β = −2.83 g/L, *p* = 0.000), and higher total protein levels (β = 2.45 g/L, *p* = 0.003) compared to the HIV/helminth uninfected controls.

## 4. Discussion

The current study aimed to investigate whether HIV and helminths coinfection worsens anaemia and malnutrition in a South African adult population. Overall HIV prevalence irrespective of helminth infection was 52%, with 15% of the population being coinfected with helminths. Approximately 33% were singly infected with helminths in the study population. The coinfected group had lower levels of vitamin A, calcium, and albumin levels compared to HIV/helminth uninfected controls. The HIV-infected group had the highest prevalence of anaemia, followed by the HIV/helminth uninfected control group, and the helminth and coinfected groups, both of which had similar anaemia prevalence.

The present study found reduced calcium and albumin levels among the HIV and helminth coinfected individuals. These analytes have a proportional relationship in blood. Calcium is a cation that is found primarily (99%) in the bone [26]. The latter contributes to the mechanical strength and acts as a calcium pool in the extracellular fluid (ECF). The remaining 1% is made up of intracellular components (0.8%) and extracellular components (0.2%). Serum calcium (part of the ECF fraction) exists in three physiological states: protein bound (40%), complexed with inorganic salts (15%), and free (50%) [27]. Albumin binds 90% of the protein-bound calcium, while globulins bind 10% [27]. Therefore, variations in serum albumin concentration change the concentration of total serum calcium [28]. Several factors may change the protein content of serum, and consequently the total calcium concentration, without affecting non-protein-bound (ionized) calcium [29]. Some of these factors may include a variety of conditions where the synthesis and regulation of serum proteins are either increased or decreased [29]. These include changes in the concentration of plasma proteins. Albumin concentrations fluctuate during inflammatory infections. This is due in part to prioritisation of acute phase proteins in the liver, in exchange for albumin synthesis [30]. The finding of lower albumin (and by association, calcium) among individuals with HIV and helminth infections in this study is therefore reasonable. 

In the present study, the most prevalent species was *Ascaris lumbricoides*, followed by *Schistosoma* spp., *Trichuris trichiura* and *hookworm* spp. Both *T. trichiura* [31] and *hookworm* spp. [32] may be a significant risk for anaemia. This is highly dependent on the severity of the infection [31], with moderate to heavy infection aggravating anaemia. The impact of both *A. lumbricoides* [33] and *Schistosoma* spp. [33] on anaemia is still unclear. HIV [9,34] and helminth [33] infections are associated with iron deficiency anaemia. Unexpectedly, in the present study, the uninfected control group had the second highest (23%) anaemia prevalence. This could be because the control group was not screened for underlying illnesses that could have influenced the study’s findings.

The HIV and helminth coinfected group was also found to have significantly higher iron levels compared to the HIV-infected group. In a previously reported study in a similar population, the opposite findings were reported [11]. As mentioned previously, *T. trichiura* and *hookworm* spp. [32,35] are the main contributors to iron deficiency. Both HIV and helminth infections have acute (early-stage infection) and chronic (late-stage infection) phases. In the present study, we did not determine whether participants had acute or chronic HIV and helminth infections, which could have confounded the results.

The reduction of micronutrients (vitamin A, calcium, and albumin) in the current study is consistent with several studies which have associated HIV [9,36] and helminth [32] infections with malnutrition. *A. lumbricoides* infection is known to intensify vitamin A deficiency [32,33]. Hookworm infections reduce food intake and increases nutrient wastage through vomiting, diarrhoea, and blood loss. These effects exacerbate protein-energy malnutrition, anaemia, and other nutrient deficiencies [32]. Whipworm treatment has been reported to improve serum albumin among helminth-infected children [32]. In addition, schistosomiasis has adverse nutritional consequences in the host similar to soil-transmitted helminths [32]. On the other hand, HIV was also associated with low micronutrient (vitamin A, zinc, and iron) levels [37,38]. These are suggested to produce adverse outcomes in HIV-infected people, including faster HIV disease progression, mortality, and increased risk of HIV transmission. The inclusion of vitamin and mineral supplements during ART have been suggested as it has been reported to improve immune status including increased CD4 counts [9,37].

The HIV and helminth coinfected group had higher total protein and lower albumin levels when compared to the HIV/helminth uninfected control group. Similar findings were made in the HIV-infected group, which showed significantly higher total protein levels and lower albumin levels. These findings were similar to those observed in a previous SA study which also demonstrated high protein levels and lower albumin levels among HIV and helminth coinfected and HIV singly infected adult populations [11]. Total protein is made up of albumin and globulin fractions, albumin accounting for approximately 60% of total protein in serum [39]. Both total protein and albumin are serum proteins produced by the liver that may be influenced by nutritional status and inflammation and infection [11,40].

Since the polyclonal stimulation of B cells in response to the acute or chronic stages of the infection and associated opportunistic infections, HIV infection causes a nonspecific expansion of the globulin fraction [41,42]. Thus, the higher total protein levels seen in both the HIV-infected group and the HIV-helminth coinfected group and lower albumin in the latter, may have resulted from prioritization in the formation acute phase proteins and/or polyclonal hyper globulin stimulation in response to HIV infection [39] and, as a result, lower albumin levels.

The majority (75%) of the participants had normal CD4 counts (>500 cells/µL) and only 5.8% were severely immunocompromised (<200 cells/µL). There was a higher proportion (66%) of participants with an undetectable HIV viral load. Lower CD4 counts [43] and a high HIV viral load [44] have been linked to severe malnutrition. However, in the current study, the majority of the participants had higher CD4 counts and suppressed HIV viral loads. This could be attributed to the efforts made through the test and treat policy [45], which appears to be working. South Africa (SA) has the world’s largest HIV epidemic; as a result, the country has an ART program that follows the “test and treat” guidelines as of September 2016 [45]. In addition to ART, the National Department of Health (NDH) promotes healthy living and proper nutrition for the general population and more so for people living with HIV [5].

The majority of the participants in the study had high MCV, which was an indication of macrocytosis (enlarged red blood cells). Vitamin B12 and folic acid (folate) deficiency are the major contributors to macrocytosis [46]. Vitamin B12 and folate are essential nutrients for erythropoiesis [47]. These nutrients are particularly crucial for the proliferation and differentiation of erythroblasts [47]. Folate or vitamin B12 deficiency prevents DNA synthesis, inhibits purine and thymidylate synthesis, and causes erythroblast apoptosis, which results in anaemia from inefficient erythropoiesis [47]. Macrocytosis may also be evident during treatment of HIV infection by zidovudine (AZT) [48]. However, in the present study, a major limitation was the lack of vitamin B12 and folic acid data, due to budgetary constraints and sample availability. Since these are mostly HIV-infected participants, a limited volume of blood could be drawn for ethical reasons. 

## 5. Conclusions

The overall findings from the study suggest that HIV and helminth coinfection may be associated with anaemia and some nutritional deficiencies. However, the present study limitations warrant that more studies that are properly designed, with interventional components, are needed. The current study’s findings highlight the importance of proper health intervention programs in addressing the burden of HIV and helminth coinfection-induced anaemia and malnutrition. The findings indicate that dual infection with HIV and helminth is common among adults living in under-resourced areas with poor sanitary conditions. More work is needed to break the cycle of coinfections and possible disease interactions. This is a neglected field of research since the majority of parasitological were reported in children; adult population studies are scare. In addition, based on our knowledge, there are little to no studies investigating the impact of HIV and helminth coinfection on anaemia and malnutrition in the adult population in South African. This has serious implications since peak HIV infections occur in adults, while helminth infections prevalence has been shown to be also common among adults. 

## Figures and Tables

**Figure 1 nutrients-14-04970-f001:**
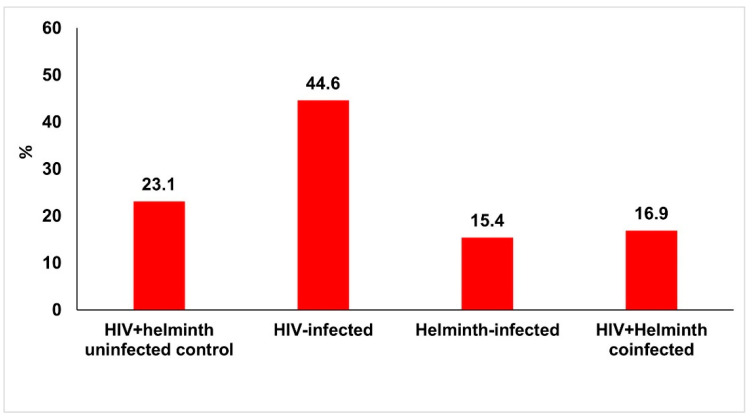
Anaemia prevalence in the study population stratified based on infection status. Anaemia was determined by haemoglobin levels of <12g/dL.

**Figure 2 nutrients-14-04970-f002:**
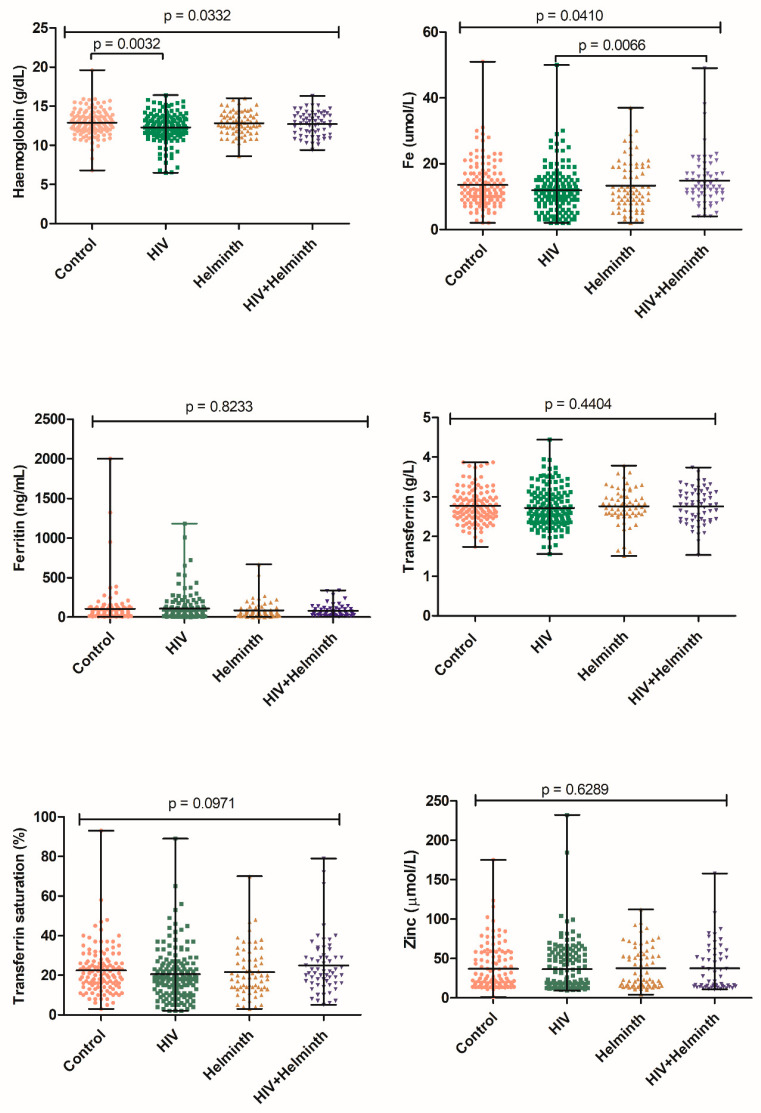
Univariate association of haemoglobin, serum iron, ferritin, transferrin, transferrin saturation, and zinc levels in HIV and helminth-infected and uninfected control groups. “Control” refers to the HIV and helminth uninfected group.

**Figure 3 nutrients-14-04970-f003:**
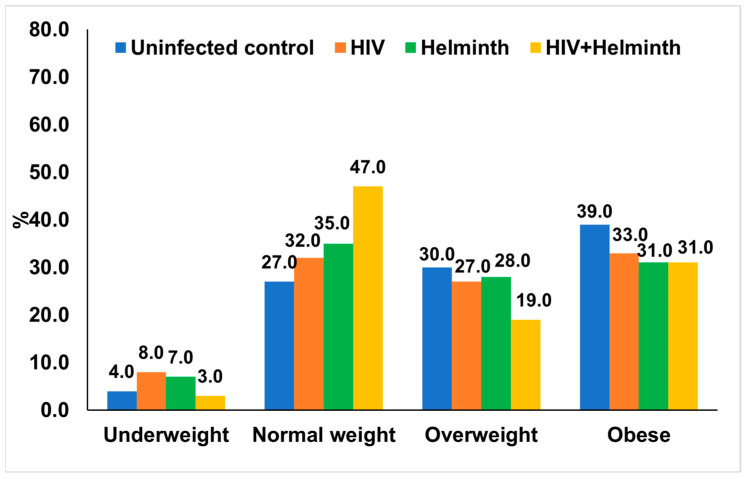
Body mass index distribution in the study population stratified based on infection status.

**Figure 4 nutrients-14-04970-f004:**
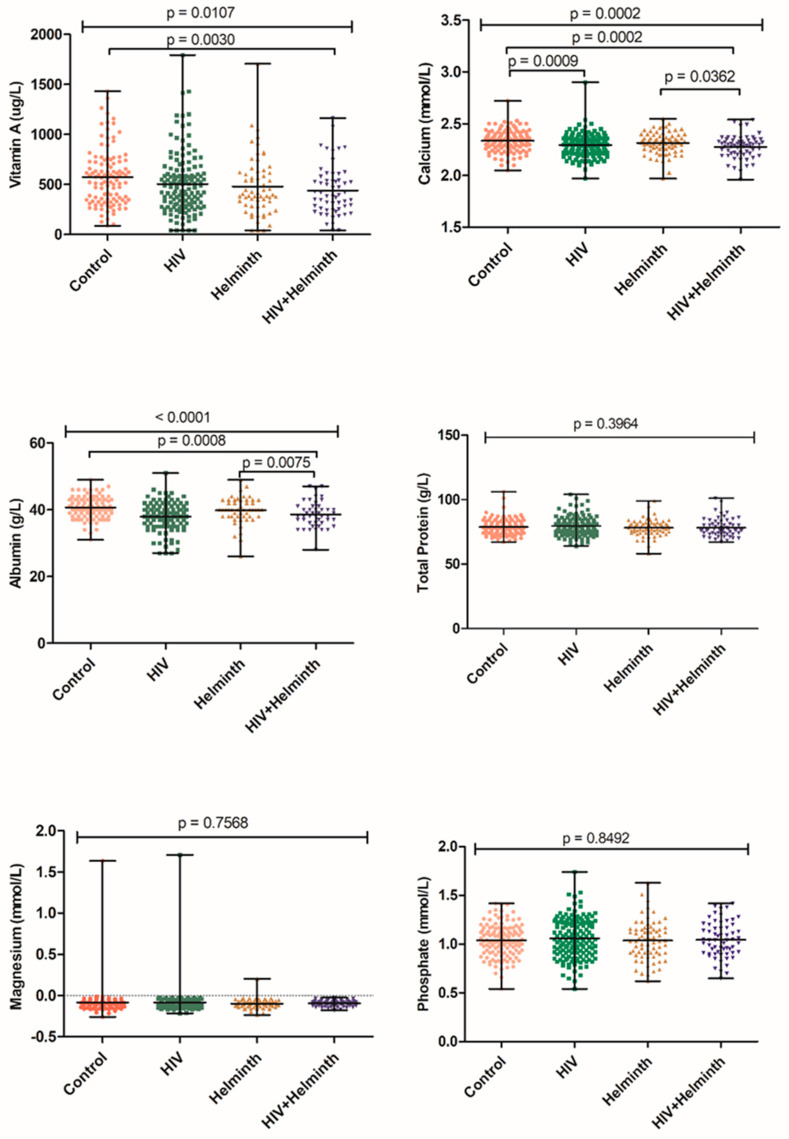
Univariate association of vitamin A, calcium, albumin, total protein, magnesium, and phosphate levels of HIV- and helminth-infected and uninfected control groups. “Control” refers to the HIV/helminth uninfected group.

**Table 1 nutrients-14-04970-t001:** Blood analytes and laboratory reference ranges.

Parameters	Reference Ranges
Haemoglobin	Male: 13.0–15.3 g/dL
	Female: 11.1–13.3 g/dL
Mean Corpuscular Volume	75.4–86.8 fL
Iron	10.0–30.0 μmol/L
Ferritin	30–400 ng/L
Transferrin	2.0–3.60 g/L
Transferrin saturation	20.0–55.0%
Calcium	2.15–2.65 mmol/L
Phosphate	0.80–1.40 mmol/L
Magnesium	0.65–1.0 mmol/L
Zinc	9.2–18.4 μmol/L
Vitamin A	200–600 μg/L
Albumin	35.0–52.0 g/L
Total protein	60–85 g/L

**Table 2 nutrients-14-04970-t002:** Clinical characteristics of all the study participants (N = 414).

Patient Characteristics			N, % (95% CI)
* HIV viral load (copies/mL) (*n* = 214)		Viral load lower than detection limit (≤20)	141, 65.89 (59.31–71.19)
		Viral load higher than detection limit (≥20)	73, 34.11 (28.09–40.69)
^#^ CD4 counts (cells/uL)		Severe immunosuppression (<200)	24, 5.8 (3.93–8.48)
		Advanced immunosuppression (200–349)	30, 7.3 (5.12–10.16)
		Mild immunosuppression (350–499)	48, 11.6 (8.86–15.04)
		Normal (>500)	310, 74.9 (70.49–78.81)
^!^ Anaemia	Males (*n* = 138)	Severe (<7 g/dL)	0, 0 (0.00–0.00)
		Moderate (7–10.9 g/dL)	3, 2.2 (0.74–6.20)
		Mild (11.0–12.9)	24, 17.4 (11.97–24.57)
		Non anaemic (>13 g/dL)	99, 71.7 (63.72–78.58)
	Females (*n* = 276)	Severe (<7 g/dL)	4, 1.45 (0.57–3.67)
		Moderate (7–9.9)	12, 4.35 (2.50–7.44)
		Mild (10–11.9)	99, 35.9 (30.44–41.69)
		Non anaemic (>12 g/dL)	159, 57.6 (51.71–63.30)
Mean corpuscular volume (fL)		Low (<75.4)	23, 5.56 (3.73–8.20)
		Normal (75.4–86.8)	45, 10.87 (8.22–14.24)
		High (>86.8)	346, 83.36 (79.7–86.80)
^$^ Iron (μmol/L)		Low (<10.0)	129, 31.2 (16.89–35.78)
		Normal (10.0–30.0)	275, 66.4 (61.74–70.80)
		High (>30)	8, 1.9 (0.98–3.77)
^$^ Ferritin (ng/mL)		Low (<30)	116, 28.0 (23.91–32.53)
		Normal (30–400)	274, 66.2 (62.50–70.57)
		High (>400)	10, 2.42 (1.32–4.39)
^$^ Transferrin (g/L)		Low (<2.0)	15, 3.6 (2.21–5.89)
		Normal (2.0–3.60)	365, 88.2 (84.70–90.93)
		High (>3.60)	16, 3.9 (2.39–6.18)
^$^ Transferrin saturation (%)		Low (<20.0)	192, 46.4 (41.62–51.19)
		Normal (20.0–55)	194, 46.4 (41.63–51.19)
		High (>55)	9, 2.2 (1.14–4.07)
^$^ Haemoglobin (g/dL)		Low (<12.0)	129, 31.2 (26.89–35.78)
		Normal (12.0–13.3)	150, 36.2 (31.75–40.97)
		High (>13.3)	133, 32.1 (27.81–36.77)
^$^ Calcium (mmol/L)		Low (<2.15)	27, 6.5 (4.52–9.32)
		Normal (2.15–2.62)	385, 93.0 (90.12–95.08)
		High (2.65)	0, 0.0 (0.00–0.00)
^$^ Vitamin A (μg/L)		Low (<200)	24, 5.8 (3.93–8.48)
		Normal (200–600)	170, 41.1 (36.43–45.86)
		High (>600)	83, 20.0 (16.48–24.17)
^$^ Albumin (g/L)		Low (<35.0)	33, 8.0 (5.73–10.98)
		Normal (35.0–52.0)	273, 65.9 (61.25–70.34)
		High (52.0)	0, 0.0 (0.00–0.91)
^$^ Zinc (μmol/L)		Low (9.2)	1, 0.2 (0.04–1.35)
		Normal (9.2–18.4)	160, 38.6 (34.08–43.42)
		High (>18.4)	226, 54.6 (49.77–59.32)
^$^ Total Protein (g/L)		Low (<60)	1, 0.2 (0.04–1.35)
		Normal (60–85)	349, 84.3 (80.48–87.49)
		High (>85)	50, 12.1 (9.28–15.57)
^$^ Magnesium (mmol/L)		Low (0.65)	8, 1.9 (0.98–3.77)
		Normal (0.65–1.10)	402, 97.1 (95.00–98.33)
		High (>1.0)	3, 0.7 (0.25–2.11)
^$^ Phosphate (mmol/L)		Low (<0.80)	32, 7.7 (5.53–10.71)
		Normal (0.80–1.40)	368, 88.9 (85.50–91.57)
		High (>1.40)	13, 3.1 (1.84–5.30)

* HIV viral load (copies/mL): classification based on the laboratory reference range. ^#^ CD4 counts (cells/uL): classification based on the WHO guidelines [23,24]. ^!^ Anaemia: classification of severity based on the WHO guidelines [25]. ^$^ Low, normal, and high ranges for each biochemical parameter were classified based on the laboratory reference ranges (https://neubergglobal.com, accessed on 5 September 2022).

**Table 3 nutrients-14-04970-t003:** Multivariate association of micro- and macro-nutrients and haemoglobin levels with HIV and helminth singly infected, coinfected and uninfected control groups.

	Unstandardized β—Coefficient Values						
Parameters	Uninfected Controls	HIV-Infected		Helminth-Infected		HIV/Helminth Coinfected	
		β	*p*-Value	β	*p*-Value	β	*p*-Value
Haemoglobin (g/dL)	Reference group	0.28	0.745	−0.24	0.242	−0.07	0.737
Iron (μmol/L)		−1.06	0.237	−1.04	0.314	1.40	0.244
Transferrin (g/L)		2.16	0.434	0.03	0.669	−0.03	0.699
Transferrin Saturation (%)		−1.25	0.436	−2.62	0.163	2.74	0.203
Ferritin (ng/mL)		−9.21	0.567	21.82	0.317	−23.69	0.113
Calcium (mmol/L)		−0.04	0.005	−0.02	0.254	−0.05	0.003
Total Protein (g/L)		2.45	0.003	−0.64	0.432	2.01	0.040
Phosphate (mmol/L)		0.02	0.388	−0.00	0.997	−0.00	0.862
Magnesium (mmol/L)		0.44	0.301	0.00	0.757	−0.00	0.850
Vitamin A (μg/L)		−18.21	0.636	−18.66	0.680	−91.44	0.040
Albumin (g/L)		−2.83	0.000	−0.74	0.197	−2.13	0.001
C-Reactive Protein (mg/L)		3.73	0.320	6.74	0.177	−0.45	0.874
Zinc (μmol/L)		0.49	0.907	−0.52	0.906	−0.11	0.951

Unstandardized β-coefficient data were adjusted for participants’ gender, age, BMI, and vitamin and mineral supplements intake.

## Data Availability

The data presented in this study are available on request from the corresponding author.

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
