# Peer review of "Anaemia and Nutritional Status during HIV and Helminth Coinfection among Adults in South Africa"

_nutrients, 2022, doi:10.3390/nu14234970_

Round 1

Reviewer 1 Report

In this study authors evaluated anaemia and nutritional status of adults with HIV and helminth coinfection in a South African adult population residing in KwaZulu-Natal (KZN).

Several points should be addressed by the authors:

Some statements are a bit misleading. For example, although it is not totally wrong to say that “HIV infection attacks and destroys the immune system cells” what in fact occurs is that the immune response generated against HIV leads to the destruction of T cells and monocytes, thus “destroying the immune system”. I would suggest a general review of the manuscript.

Please explain how was calculated the total number of participants (n = 414 participants). This is also important considering the following statement at the Discussion: “This could be because the control group was not screened for underlying illnesses that could have influenced the study's findings.” What were the inclusion/exclusion criteria for this research?

Since authors claim that KZN “has eleven primary health care clinics (PHC)”, please explain how and why five were selected.

Authors claim that different water sources were available to people as were different toilet structures. Since both factors can be important concerning helminth exposure and infection, please present more detailed data about such conditions.

According to the Introduction, South Africa follows "test and treat" guidelines, thus I inferred that all HIV+ participants of the present study are under ART. Nevertheless, authors claim that “This detection limit was used to further categorize those who were or were not receiving antiretroviral therapy (ART).” This point deserves a better explanation.

In fact, I believe that HIV+ patients should be separately evaluated considering ART treatment status.

Please explain data from Table 2 (it is quite confusing to identify if data came only from HIV+ patients or from all participants). Please clarify this point.

 Also at Table 2, if “Anemia, Male, Severe” patients equal to zero, how “0, 0 (0.00-2.71)” is listed on column concerning “N, % (95%CI)”? Revise Table2.

In fact, I would prefer a table 2 comparing data from groups a) HIV+, b) helminth infected, c) co-infected, and d) healthy (not HIV and/or helminthic infection).

Figures 2, 3, 4 and 5 depict such classification, although instead of “control” I would call such group as “healthy (not HIV and/or helminthic infection)”

At the Discussion, please define what author’s means by “acute or chronic HIV and helminth infections”.

I was puzzled by one of the so called limitation of the study “This resulted in having more than half of the participants being HIV infected. This could have skewed the data.” Since, as far as I understood, the aim of the study was to evaluated anaemia and nutritional status of adults with HIV and helminth coinfection. Therefore, HIV positivity should be not seen as a limitation. Please explain.

As “11.4% of the total population were taking supplements”, I strongly suggest to exclude such patients (ora t least to evaluate data separately).

Minor points

There are several typos and grammar errors (for example “coroscopy” instead of “coproscopy”; “…This is due to in part to prioritisation...”

Author Response

Dear reviewer 

Reviewer 2 Report

There are several areas in the manuscript that deserve improvements:

1. statistical methods; the dataset should be analyzed  with multivariate analysis since anaemia and nutritional  status are influenced by demographic variables as age and  sex.

2. Introduction should be limited to the purpose of the work and its significance and should  review the current state of the research field .

3. Improve the explanation of the percentages reported in figure 4.

4. Additional data should be included vit. B12 and folate and MCV 

Author Response

Dear reviewer 

Round 2

Reviewer 2 Report

Dear Authors,

    I have studied the rebuttal and revised version of your manuscript and I still find it lacking. Major points to considered are:

-        The diagnostic criteria for helminth infections have not been adequately provided. In particular the diagnosis of Ascariasis, which is reported as the most prevalent infection in this cohort, is based on serology which cannot be considered reliable, unless siero-conversion is demonstrated. Moreover, serological cross-reactivity with other infection and atopic disease (HDM allergy) has not been taken into account.

-        The reference range for anemia has not been differentiated for male and female population thereby resulting in an overestimate of anemia in the female population

-        The high percentage of red blood cell macrocytosis and the lack of vitamin B12 and folate determinations have not been even discussed and represent one of the major limitations of the study.

If these issues cannot be addressed I find it difficult to have this study published.

Author Response

Dear Reviewer 

Please see the attachment below.

Regards,

MM

Round 3

Reviewer 2 Report

We appreciate the effort of the Authors to improve the manuscript.